# Neomythology: A New Religious Mythology

**Ioannis Xidakis** 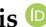

Independent Scholar, 11471 Athens, Greece; ixydakis@gmail.com

**Abstract:** The purpose of this paper was to present the link between the myths in the plots of modern entertainment products, such as science fiction novels, movies, comic books and video games, with the motifs of New Religiosity. The plots employed in these products are of a mythical nature. This re-emergence of myth is once more taking centre stage today. The social sciences and humanities concerned with this phenomenon call it "neomythology". Neomythology, in whatever form, as a comic book, video game, etc., shows gods and supernatural heroes or villains interacting with one another in certain ways. The ways these mythical elements are combined and displayed confirm that neomythology belongs to New Religiosity and repeats motifs of New Religions and New Age myths, such as the secularization of the absolute, that is to say, the placing of the transcendental element in the material and explainable world, the absolutisation not only of the acquisition of power but also those who bear it, the correlation of magic and technology, and the reuse and utilization of the mythology of the past and of traditional religious teachings, in order to produce a new myth.

**Keywords:** modern mythology; new religiosity; Western culture; secularization; technology; magic

---

## 1. Introduction

In order to examine neomythology, namely, the contemporary nature of myth as it develops within the New Religiosity, and to approach the subject of the presence of myths in modern recreational products, we need to answer the following questions: "What is myth"? "What is the content of myth"? Since the 19th century, however, whenever attempts were made to answer these questions, myth or mythical stories as a whole, in other words, mythology, was associated with fantasy, religion and past civilisations.

More specifically, in the 19th century, under the influence of the Enlightenment, myth was viewed as the opposite of reason and reality. Enlightenment thinkers tended either to dismiss myth as an irrational element that was incompatible with science or to associate it with childish expression or the infancy of civilisation (Segal 2004, p. 3; Bristow 2017; Coupe 2009; Cupitt 1952, p. 12).

In the late 19th century, in his theory on cultural survivals, Tylor [1871] (Tylor [1871] 1920, pp. 112, 113, 133, 285, 477, 501) regarded myth as a part of a culture that belonged exclusively to the past, or that had survived from the cultural past of animistic religion, the era of occult science, divination and astrology. Mythical narratives as a whole belonged to the distant era of, as he put it, "primitive biology" (Tylor [1871] 1920, p. xii), "primitive psychology" (Tylor [1871] 1920, p. 471) and "childlike science" (Tylor [1871] 1920, p. 332).

Tylor identified the content of myth with excess, misinterpretation and the distortion of reality (Tylor [1871] 1920, p. 368). At the same time, however, he rejected any attempt to interpret it symbolically or metaphorically or to regard it as moral history. Thus, he treated myth as a form of science, and he connected it with the attempts by a primitive society to interpret and describe what it believed to be real (Tylor [1871] 1920, pp. 299, 316, 409; Segal 2004, p. 19).

In other words, Tylor disassociated myth from modern life, just as the theorists of the Enlightenment did, while at the same time, through his close study of it, he concluded that in the past, it was a real and important factor in people's lives. Despite its cloak of fantasy,

myth is based on reality, is inspired by it and is perceived as a real fact (Tylor [1871] 1920, pp. 299, 300, 316, 317, 409, 445).

In the 20th century, though he restricted myth to the past, the anthropologist Claude Levi-Strauss (1985, pp. 110, 114, l27, 146, l57, 158, 159, 161, 190, 191) presented it, just like Tylor, as a body of stories that played an important role in the lives of primitive communities. The members of those societies used myths to define certain rules of action and constructed a homogeneous system that attempted to establish their racial origins, the existence of their ritual practices and their system of social organisation all at the same time.

The theologian and New Testament interpreter Rudolf Bultmann associated myth with the world view of modern society. In this association, and in his comparison of myth with science and theology, he rated it as an inferior cultural element. For Bultmann himself, however, mythology can also contain truths since it has always had the power to refer to strange, transcendental and mysterious phenomena and give worldly objectivity to unworldly or divine things. Therefore, in this sense, myth is important and functions as a vessel of truths (Hege 2017, p. 46). For the truth of a myth—or, as Bultmann says, the kerygma—to be revealed, a myth must be interpreted and its content must be made to reveal the world of God and the entirely other (McLaughlin and Smid 1999, p. 19; Congdon 2015, p. xxix).

Bultmann's contribution to the perspective of myth, within the framework of dialectical theology, is precisely this: he connected myth with history, although at the same time, through the process of demythologisation, he removed it from its original cultural context in order to stress its diachronic content, which might conceal the divine truth. In this way, myth came to occupy a special place since it may convey messages about the very existence and nature of man (Congdon 2015, pp. xxvi, xxx).

In the second half of the 20th century, Mircea Eliade presented a myth as a sacred history, the history of archaic men in relation to the supernatural beings and the way in which the latter created and presented the world, time and history. Myth becomes the language used by archaic society to bring the sacredness of its gods into the present and into its own reality, that is, profane space and time (Coupe 2009, p. 53; Eliade 1963, pp. 5, 6).

Eliade agreed with Tylor that, for ancient civilisations, myth revealed not only a moral history but a real event (Eliade 1963, p. 6). This event was not merely a past event like many others; on the contrary, as a sacred event, it was a paradigm for everyday life. By emphasising the sacredness of myth in this way, Eliade was in agreement with Bultmann. Just as the latter believed that myth contained divine truth, Eliade held that for those who believe in a myth, it has a transformative effect as a special religious experience. The religiosity of myth leads its followers into the world of the supernatural presence of the gods (Eliade 1963, p. 18).

The remarkable and special thing about Eliade (1961, pp. 11, 16, 18, 19, 25), however, is that he did not view myth only in terms of the past or any ideal content it may have, but he also connected it strongly with the present. He discovered that not only has myth not been edged out or even, for that matter, eradicated from modern life but it is also in fact present in its different aspects. The myths of the past, which once expressed hierophanies, that is to say, the presence of God in men's lives, continue to fascinate modern man. In this sense, they have turned into new myths and have thus survived in modern literary products, dreamy and fanciful images, and symbols.

The literary theorist Kenneth Burke (1966, pp. 18, 20, 21, 51, 212, 381, 384–386) made an interesting observation: myth is based on real-life situations, which, after a key moment in history, became models, that is to say, mythical stories that served as examples, as a timeless paradigm for the community. Thus, myth promotes the idea of perfection, perfect things and stable principles, such as the duo of "good and evil". The body of mythical stories speaks about the beginning of time, the origins of man, the birth of the gods, the beginning of a ritual, the perfect hero and the perfect good or the perfect evil.

For the theorist of hermeneutic phenomenology Paul Ricoeur, too, myth should be approached from a different angle. Ricoeur's (1969, p. 6) perspective coincided with that of

the scholars mentioned above, and above all, the approach adopted by Bultmann, Burke and Eliade, who held that myth is not merely a false interpretation of the world. On the contrary, as Bultmann stressed, it reveals secrets about man's destiny and nature. It reinforces the development of a moral theory and man's perception of good and evil, a view which, to a certain extent, can also be found in Burke's work (1969, p. 9). It expresses its relationship with what is regarded as sacred, thus recalling Eliade's (1969, p. 5) worldview.

For Ricouer, traditional myth, as a symbol that has lost contact with its origins in time, becomes an important dimension of modern thought (Ricouer 1969, p. 5). It serves as a constant motivation to modern man to look beyond the narrow confines of his reality, of his daily temporal and spatial constraints, to another, possible world (Ricouer 1969, p. 8).

For contemporary mythologists, it is clear that myth does not belong exclusively to the past. Contemporary culture is full of mythical representations and mythical symbolisms.

To be precise, Laurence Coupe stressed the point that myths are repeated today in various forms of entertainment, such as works of literature and cinema films (Coupe 2009, pp. 100, 191, 192, 199). As a result of this presence, myth plays an important cultural role (Coupe 2009, p. 112). Anthony R. Mills recognised the fact that the monomyth, as he calls it, of the 19th-century American hero who saves the community and leads it into a new era is reproduced in a variety of modern mythical versions, such as the plots in comic books that are full of superheroes (Mills 2014, pp. 4, 22, 27, 53, 169). John Izod (2003, pp. 1, 15, 35, 80) recognised the phenomenon of the reinterpretation and reuse of mythical schemata in the products of the big and small screens. Roland Barthes presented the myth as a type of speech; as a system of communication; and as a message that conveys true and, at the same time, unrealistic content. With this dynamic, the myth functions as a second language, as a meta-language (Barthes n.d., pp. 1, 3, 40).

Myth, therefore, as a product of the human spirit, is indeed a diachronic combination of fantasy and reality, that is to say, of irrational and rational elements. In this sense, in modern entertainment and amusement products, such as video games (Xidakis 2018) and the cinema, it is possible to see the dominance of the mythical element. The plots employed in these products are of a mythical character. Precisely because this re-emergence of myth is such a pronounced phenomenon nowadays, it has been termed "neomythology" (Hexham and Poewe 1997, pp. 92–94, 152–154; Rothstein 2006, p. 649; Hanegraaff 1999, pp. 153, 158). This term is used in different disciplines dealing with the phenomenon, including that of religious studies, which, among other things, studies the phenomenon of New Religiosity.

The purpose of this article was precisely to highlight this contemporary mythical space, which develops and reproduces specific motifs of the diffused new religiosity[1].

The stages of our research path, which also underlined our purpose, were the following:

a. Initially, a definition of New Religiosity was provided and its main characteristics were analysed.

b. Thereafter, once the characteristics of New Religiosity were identified and described, a definition of neomythology was provided. Through identifying and describing the new religious motifs and defining the concept of neomythology (i.e., contemporary mythology), we aimed to show how the latter reproduces motifs of the diffused New Religiosity.

c. Consequently, this also constituted the core of our efforts and the very essence of our purpose: we aimed to identify and highlight the various aspects and meanings that each of these new religious motifs assumes in the plots of video games, comics and fantasy films.

In order to fulfil our purpose, our method moved in three directions. Our method was as follows: (a) systematic, in that we grouped the main characteristics of New Religiosity that emerge from our study into seven basic motifs; (b) analytical, since we analysed the repeated themes and repeated concepts arising from the correlation between the diffused new religiosity and neomythology and then we identified the new religious motifs in the plots of neomythological products; and (c) historical, mainly with respect to the examples of neomythology identified, which were organised on the basis of their historical development and formation. In other words, we provide examples of the new religious schemata we

identified from contemporary neomythological products on the basis of the order in which they appeared from the 1980s up until the present day.

In the pages that follow, and based on the purpose and methodological framework we set, we approach and analyse our position: myth does not belong exclusively to the past. The 20th- and 21st-century civilisation is full of mythical representations and mythical symbolisms. The plots of modern leisure and entertainment products introduce their users, as viewers or players, into a universe where realistic elements coexist alongside fantastical ones. In fact, the way in which these elements coexist and the motifs that emerge from this coexistence, and from the co-inherence of the plausible and the implausible, all point to the main characteristics of a new mythology within the limits of New Religiosity.

## 2. New Religiosity

The term "New Religiosity" refers to non-traditional modern religiosity, that is to say, the type of religiosity that has been formed through the teachings and practices of the New Religions.

The New Religions made their appearance on the religious map mainly after the 1960s (Papalexandropoulos 1997, p. 153; Campbell 2001, p. 37; Barker 2001, pp. 15, 16; Chryssides 2001, p. 257; Cowan and Bromley 2015, pp. 6, 7; Bromley and Melton 2004, pp. 42–43), although their characteristics are evident even in various religious groups of the 19th century (Papalexandropoulos 1997, p. 149; Hick 2004, p. 307; Hexham and Poewe 1997, p. 70; Cowan and Bromley 2015, p. 6). They are called "new" for various reasons, including the following: (a) In terms of their appearance and development, they are associated with what is new, with what is "modern", with modernity and postmodernity, that is to say, with all the changes that have taken place in the Western world since the 19th century (modernity) and the 20th century (postmodernity). These changes have been produced by a variety of factors, one of the most basic of which, of course, is industrialisation, with the radical social, economic and technological changes that it brought in its train (Hexham and Poewe 1997, pp. 36, 37, 147–151). (b) They differ from traditional religions (e.g., Christianity), that is to say, from the religions that, until the ages of modernity and postmodernity, traditionally dominated every aspect of society. Schematically, it could be said that on a virtual world map, the New Religions lie in the gaps that appear between the spheres of influence of the traditional religions.

## 3. The Distinctive Features of the New Religiosity

The New Religiosity that has developed since the emergence of the New Religions (or New Religious Movements) and as a result of the latter's activities is characterised by both the fluidity and mobility of (a) organisations; (b) teachings; and (c) to a certain extent, of religious followers too since these move between one group and another, investigating the strategy, principles and teachings of each group in turn, without necessarily being bound by only one view of the "truth" about the world. We say "to a certain extent", because the mobility of the followers characterizes certain areas of the New Religiosity, mainly that of the New Age. Along with the phenomenon of mobility, there is also the fanatical dedication of the follower, especially when they belong to New Religions, which are organizations, such as Scientology.

In other words, the New Religiosity is an alternative, disparate religious reality that appropriates everything: from psychology and medicine to ecology, from gnosticism to interplanetary travel, from the ancient myths about gods and heroes to modern myths about aliens, and from magic to technology.

All of these elements, depending on which new religious group adopts them, are presented either in the most popularised and over-simplified way or cloaked in a mantle of philosophical or scientific knowledge, or even as a combination of both of these.

In general terms, the distinctive features of the New Religiosity that emerge from a study of the diverse teachings of the various New Religions are as follows:

(a) The broadened concept of human potentialities. Man, both collectively as a human race and as an individual, finds himself called upon to exploit his many potentialities, to reach a state of self-actualisation (Hexham and Poewe 1997, p. 148; Heelas 2001, pp. 54, 60), self-expression (Hexham and Poewe 1997, p. 155), self-enhancement (Heelas 2001, pp. 54, 60), etc.

(b) The conviction that there are two levels of reality: (i) the external, everyday reality and (ii) the internal reality of the consciousness. The latter is superior to the former (Heelas 2001, pp. 52, 53).

(c) The conviction that what human beings have become accustomed to seeing as "part" of reality is in fact the visible, outward part of it. On the other hand, the "whole" and the "one", the holistic approach to life and the unity of the relationship between God (as the Absolute) and man, and the unity of the universe, all religions and mankind, constitute the essence of reality, its real side (Papalexandropoulos 1992, p. 45; Campbell 2001, p. 40; Heelas 2001, pp. 52, 60; Hexham and Poewe 1997, pp. 5, 147; Sutcliffe 2003, p. 11).

(d) The conviction that an awakened person, who has recognised and accepted this holistic understanding of the world, will accelerate change: through their awakening, they will bring spiritual beings and alien leaders to the forefront of history; they will usher in a new social, technological, planetary and ecological future (Xidakis 2006, p. 8; Sutcliffe 2003, p. 21).

(e) The dominant idea, as a result of the foregoing, of a transformative vision of another world, of a new space and time, of a New Age for the world and for man (Papalexandropoulos 1991, pp. 17, 19).

The term "New Age", in particular, has a range of different meanings within the New Religiosity. On this point, the following should be noted.

First of all, the term "New Age", although it generally occurs in the teachings of the 20th-century new religions and is even used in connection with certain groups in the last decade of the 19th century (e.g., the Theosophical Society), denotes the New Age religious movement, which made its presence felt after 1960 and which forms a subdivision of the New Religiosity (Papalexandropoulos 1991, pp. 20, 21; Xidakis 2006, p. 7; Barker 2001, p. 17; Sutcliffe 2003, pp. 1–6, 11, 21, 29, 30, 198). Its main characteristic is that its followers are not bound by a defined set of practices, do not belong to any one particular group and do not rally around a central leader (Sutcliffe 2003, pp. 198, 224; Barker 2001, p. 17), as happens in the New Religions, which are based on the preaching of a charismatic leader (Barker 2001, p. 20; Clarke 2006, p. viii).

Second, the process of seeking and striving for the New Age, which is very often a diffuse and porous one, involves striving to achieve a variety of different goals: New Age followers are said to embrace the notion of a mankind with limitless capabilities and of man as a traveller of the Universe (Papalexandropoulos 1992, p. 52); they are said to believe in doing away with the bounds of daily, conventional and traditional religious and social experience, and to believe in the salvation and healing of the soul, the tranquillity of the mind, the improvement of karma and a state of bliss after death (Papalexandropoulos 1991, p. 20; Wilson 2001, p. 7), and even professional success, personal and financial fulfilment, and the acquisition of luxury goods (Heelas 2001, pp. 53, 66; Sutcliffe 2003, pp. 29, 30; Clarke 2006, p. ix).

At this point, it is worth pointing out that this tendency towards the worldly, that is to say, New Ager's preoccupation with improving their professional and financial status, resembles the desire held by the heroes of the products of neomythology (e.g., science fiction films and novels and video games) to have more possessions, weapons and even qualities.

(f) The tendency for the history of human civilisation, the mythology of the past and religious teachings to be utilised by each group, or each group leader or member, in a new way (Hexham and Poewe 1997, pp. 94, 95, 150, 153; Cowan and Bromley 2015, p. 9; Xidakis 2006, pp. 19, 21, 27, 36, 42, 53–54, 59–60, 78–89).

(g) The tendency for the traditional to be dismissed as inadequate. In this context, traditional religiosity is rejected as being restrictive (Clarke 2006, p. ix; Cowan and Bromley



2015, p. 9; Hanegraaff 1996, p. 88; 1999, p. 152), conventional science is dismissed as being incapable of providing answers or even dangerous (Hexham and Poewe 1997, pp. 93, 149, 150) and conventional knowledge of history is rejected as being indifferent (Hexham and Poewe 1997, p. 150).

This last motif in particular—the questioning of history—follows the previous one, that of the reformulation of history, since the refusal of the adherent to a New Religious Movement to accept history "gives them the freedom" to interpret it from the very beginning, to rewrite it and to adapt it to their own worldview.

Here I would like to make a few observations.

First, when I concerned myself with the alien myth and the new religious groups that attach religious significance to aliens, I studied Erich von Däniken's view of history. His view of the world and, above all, the evolution of the history of civilisation is a typical example of how these two motifs function and are interconnected.

Erich von Däniken, then, was a supporter of the alien myth, and also a supporter of a similar myth about ancient alien astronauts. Von Däniken had his own view of the ancient past and especially of the evolution of civilisation, and within the spirit of this freedom to tell his own "human story", he claimed that the ancient myths about superhuman beings, gods, demigods, heroes and monsters do not constitute an illusory and imaginary history. On the contrary, in his view, they constitute a legacy of the real knowledge of the past to the future, a reliable and essential complement to real history. It is only through the ancient myths that the real character of recent and modern history can be revealed. In other words, von Däniken rewrote the history of mankind from the very beginning, while refusing to recognise the official version that we know and have been taught on the grounds that it is deficient and dangerous (Von Däniken 1969, 2009, 2011).

Of course, von Däniken was not the first person to do so. A similar tendency to reinterpret history can be found in new religious groups of both the 19th and 20th centuries. For example, in the 1880s, Helena Blavatsky promoted her views on the ancient world, while in the 1970s, Claude Vorilhon, who founded the new religious group the Raëlian Church in 1973, promoted his. The differences between their views and von Däniken's lie in the fact that Blavatsky's view of "ancient history" formed part of her teaching on the Ascended Masters and the Great White Brotherhood (Blavatsky 1888a, 1888b), while Vorilhon associated similar views with his belief in the existence of the Elohim, a culturally advanced race of aliens (Vorilhon 1989). Also, even in the sphere of diffused religiosity, von Däniken is not the only example of a writer attempting to present a new or different history. For example, Shirley MacLaine, in her theory of the "Inner Christ" (Christ Consciousness), also attempted to make such a reformulation of what is historically acceptable (MacLaine 1988, p. 208; 1983).

In particular, this freedom in reinterpreting history and in promoting another, alternative side of it is strongly evident in the plots of the entertainment products of contemporary popular culture. It is precisely these products in the form of cartoons, movies, video games and books that constitute the domain of neomythology.

## 4. The Distinctive Features of Neomythology

The term "neomythology" defines the modern tendency for symbols and the religious and mythical figures of world history to be removed from their natural context or religion and to be reinterpreted, thus "constructing" a new myth. In this sense and, in particular, as a fantasy element in the plots of the entertainment products of contemporary Western civilisation, neomythology forms a part of diffused religiosity, which, in turn, forms a part of the New Religiosity. More specifically, diffused religiosity is that kind of religiosity that does not take the form of movements, groups or organisations. On the contrary, it is characterised by the free movement of ideas and practices expressed in various media, such as television or art. The neomythological schemata are influenced by the mythology of the New Religions and diffused religiosity and reproduce their basic distinctive features (Xidakis 2021).

As far as the fluidity and inclusiveness of this neomythological space allows, the main motifs of this contemporary mythology, that is, the mythology that develops through the plots of entertainment products of the contemporary Western world, such as science fiction movies, comics and video games, are as follows.

The integration of the absolute, i.e., the divine that various neomythological products include in their plots, in the boundaries of the world. Under no condition does the divine constitute something different as well, where different refers to something that is completely outside of history, matter and the world. As a result of this perspective, the supernatural—in the sense of magic or divine intervention—is objectified, namely, used as a means to an end or very simply as a tool of salvation or destruction.

Based on the previous motif, the image of the supernatural hero is promoted, who—after recognising and accepting the divine element in the world and its aspects—will effect change: through their actions, the hero will discover new knowledge or reveal the secrets of a new science, they will create a new civilisation and a new technological era for the mythological world. It thus dominates alongside the motif of the chosen hero and the vision of another world, of a new space and time, of a new historical period, of a new science.

Particularly, with regard to this new version of science, we see that in neomythology, and generally in the diffused religiosity, complex and multilateral correlations between the magical and scientific elements develop. Magic and technology in the neomythological context form the amalgam of magical technology or scientific magic. This means that the magical skills of a neomythological character are presented as explainable powers. On the other hand, technology either has an otherworldly, divine or alien origin; or enhances its user's skills; or even identifies with magic, in the sense of the automatic fulfilment of their will. It is incorporated, in turn, in the same plausible and convincing framework as magic, thus shaping the characteristics of a supertechnology.

Therefore, within this multi-prismatic space of neomythology, each aspect, such as the understanding of science, the traditional religiosity and mythology, and the history of civilisation, is used each time in a new way. To put it differently, everything is determined from the start and is adapted depending on the plot of each contemporary mythological product. Especially, as discussed further down, the history of civilisation, as a sequence of events, along with everything it contains, namely, its whole set of myths, legends and traditions, becomes the material for telling another, alternative version thereof, through which a different history of humankind is revealed.

Neal Stephenson's novel *Snow Crash* is one of the most characteristic examples of this neomythology, as in this work of science fiction, Stephenson makes use of the term "avatar". This term later became one of the basic features of the fantasy plots of video games, the most modern and popular products of neomythology.

In *Snow Crash*, then, Stephenson (1992, pp. 222, 223, 241, 242, 342) relates the adventures of Hiro and describes the ways in which a virtual world, the Metaverse, functions. Hiro is a "hacker", a superbly skilled player and operator of the Metaverse. In this capacity, Hiro learns the following: (a) The ancient Sumerians were the first to possess supertechnology and computers. (b) Sumerian computers were able to alter man's state of consciousness. (c) Language had the power either to help men evolve or to make them ill, just as it had with computers. In other words, language functioned as a programming language, as computer code. (d) The ancient gods were the scientists and technologists of the past (Stephenson 1992, pp. 342, 352). (e) There are both good and evil demons; good demons bestow good health, while evil ones cause confusion and biological, emotional and computer diseases (Stephenson 1992, p. 200). (f) Technology brings power and functions like magic. Indeed, its new initiates—hackers and those entering the Metaverse, the avatars (digital representations of real people)—are capable of creating a new world and controlling it through the power of these magic computer languages (Stephenson 1992, p. 351).

*Snow Crash* reproduces conceptual schemes and motifs of the New Religiosity and neomythology through the following: (a) The hero who, as the good guy and chosen one, attains a state of self-actualisation and awakening. (b) The activity of the hacker-avatars,

who introduce a new age into real time. (c) The hackers and the ancient gods, who are also technologists—that is to say, they are technically upgraded—and magicians; in other words, they succeed in evolving consciousness and in attaining superior, arcane knowledge.

Evidently, *Snow Crash* is not the only typical example of neomythology. In comics, such as those produced by Marvel and DC, which are full of myths and adventures of superheroes, similar elements may be recognised: (a) Gods that owe their power to their supertechnology, such as the Gods of Asgard, who share their world with elves, giants, demons (Teitelbaum and Forbeck 2014, pp. 148–49) and Creator Gods, such as Apocalypse, the god of ancient civilisations (Sanderson and Forbeck 2014a, p. 25), or Gaea, otherwise known as Mother Earth, who is presented as the embodiment of life, vegetation and the cycle of nature (Brevoort and Forbeck 2014, p. 142). In Gaea, we can also see a combination of neomythology and ecology, which represents a typical theme in the sphere of diffused religiosity. (b) Characters that take the form of angels, such as the alien angels known as the Thanagarians (Wiacek 2016a, p. 308), or demons, such as Darkseid (Wiacek 2016b, p. 82). (c) Technologists, such as Batman (Manning 2016a, pp. 28–30) and the scientist Mobious (Manning 2016b, p. 15), and also magicians, such as Doctor Strange (Sanderson and Forbeck 2014b, p. 112). (d) Characters that can bring about the end of the world or save it from this fate and lead it into a new era, or that can bring about both, depending on the circumstances, as happens in the case of Superman (Eco and Chilton 1972; Reynolds 2013, pp. 102–104; Mills 2014, pp. 22, 27) and Death. The latter is an entity that sometimes seeks to destroy the universe and at others complements the work of the embodiment of life, of Eternity (Darling and Forbeck 2014, p. 104; "Death (Earth-616)" 2021).

It is particularly this fluidity with regard to who is good and who is evil that characterises the diffused religiosity that diverts itself by redefining historical figures and the history of the past. A typical example of this in superhero comic book mythology is the case of Doctor Mid-Nite. In the mythological world of DC Comics, he is initially presented as a hero and arbiter of justice. With his superhuman ability to see in the dark, he hunts down drug dealers in the 1940s and 1950s. However, his upgraded status to a crime fighter later leads him to become a member of a parastatal group of superheroes, the Justice Society of America (Cowsill 2016, p. 92).

In science fiction series, such as *Star Trek* (1966), immaterial and incorporeal beings appear, such as the Q, which influence the course of the Earth's or human history and manage the properties of space-time, and also a variety of heroes and antagonists who can "beam themselves up", as the phrase goes, meaning they can teletransport themselves with the mediation of supertechnology, or as a result of their genetic or intellectual evolution ("Transporter" 2021; "List of Star Trek Aliens" 2021[2]; Hexham and Poewe 1997, p. 84).

"New self", "supertechnology" and "god-like figures" are the basic elements that characterise the virtual universe in *Altered Carbon* (2018). According to the plot of this science fiction series: technology ensures the evolution of the self, offering the magical possibility of immortality, and all eternal, immortal and superhuman elements are presented as being scientific and rational, even if they call to mind terms, concepts and names used in the mythologies of the past (for example, the class of the immortal nouveau riche is identified as the class of the "Methuselahs" or "Meths") ("Altered Carbon" 2021).

The plot of *The Matrix* (1999) is typical of science fiction films, and also impressive for the way in which it presents a new myth as a reuse of traditional mythical schemata. In an apocalyptic future in which machines have triumphed and have imprisoned the minds of humans in a virtual reality (the Matrix), the prophetess (The Oracle) and her pupil (Morpheus) discover the saviour of mankind, Neo (The One), in this virtual realm. Neo controls the technology used by the machines in an almost magical way: with the aid of Trinity, his female companion, with whom he now forms a holy couple, he wages war against the machines and his demonic adversary, Agent Smith, and in the end obeys the creator of the virtual world, the Architect, and sacrifices himself in order to save the human city, the holy city of Zion, thus ushering in a new era for both men and machines.

The plot of *The Matrix* reveals an interesting mythical reality in which the following occurs: (a) The dividing line between good and evil exists, that is to say, there are "good men" and "evil machines", but at the same time, this line does not seem to be completely clear. (b) The heroes and antagonists, as supernatural beings, fight over the world while they are in the world; in other words, they are not transcendental or metaphysical beings. (c) Technology functions like magic and in this form, that is to say, as magical technology, is used in a new, always plausible way. (d) The mythological element remains strong and, as there is talk of a saviour (or rather, as the Architect reveals, a series of saviours) and a Creator (the Architect) of the virtual city, there is also a strong religious element. Myth and religion, however, are presented with new content. (e) All these fantastical, magical and religious elements are presented in a context of computers and machine domination; hence the hero-saviour is described by the Architect as a "program" that reloads the Matrix, the craft in which the hero attempts to attack Machine City is called "Logos", and the machine in which he finally surrenders his dead body is called "Deus ex Machina" ("Matrix" 2021; Coupe 2009, pp. 191–92).

*The Matrix*, then, reproduces many of the schemata that are to be found in traditional and modern mythologies. At the same time, however, through a combination of rational and irrational elements (as mentioned earlier), it also produces a contemporary myth.

Video games, too, in order to construct their plots, use symbols that are familiar from religious traditions and the mythological wealth of the past. They repeat themes that are familiar from the New Religiosity and neomythology. This repetition concerns not only peripheral, minor parts of their scripts but mainly their central part, which concerns the relationship between the hero and their adversary, which, in turn, becomes another aspect of the relationship between good and evil.

In the video game *Bayonetta* (PlatinumGames, Sega 2009), we not only have a repetition of elements from classical theology and mythology but also a new mythology. The plot leads to the following conclusions: (a) The demonic witch Bayonetta is good because she saves the world from the evil angels as the chosen one, the best of the witches. (b) Bayonetta awakens to her identity and upgrades her abilities and, indeed, as in the sphere of New Religiosity, this element confirms her moral superiority and her goodness. (c) The game's virtual characters—demons and angels—live in a higher sphere of existence, though always in the material world. In other words, the good guys and the bad guys in the script, just like the supernatural beings in *The Matrix*, are presented as living in the material world and are imaginable and credible characters. For example, as they fight, the demons and angels upgrade their abilities and weapons and appear to dwell on planetary dimensions, such as Paradiso, Inferno, Earth and Purgatorio, to possess a hierarchical structure, and to use technology and weapons, amongst other things. (d) There is no difference between magic and technology, but rather—as previously stated—both form two sides of the same coin, that of magical technology or scientific magic. (e) Like the follower of a new religion, the heroine recognises that all things are united with everything else since; unlike ordinary digital people, she can discern angels and demons in the full light of day and in the virtual city, while she knows that Paradiso and Inferno are linked by a network of passages. Once again, as in the New Religiosity, this is a sign confirming her select status ("Bayonetta" 2021).

In the adventure game *Darksiders* (THQ 2010), Heaven and Hell do not appear as places or states occupied by saints or sinners. They constitute two rival kingdoms (the Kingdoms of Heaven and Hell) that fight against each other to gain control over Creation. Between these two worlds of Angels and Demons, the Creator first placed the Charred Council and then later the Kingdom of Man to secure a balance in Creation ("Darksiders" 2021).

In this way, the game reproduces a motif familiar in diffused religiosity: that of the abstract divinity, which is identified with Good. According to this motif, the Creator, as the supreme god, is the only good power in virtual reality. He is an impersonal god who does not reveal his emotions to the digital characters. He stands outside the reality of the war they are waging, he does not interfere in the conflicts between the hero and the antagonist

and supports neither one nor the other, and he exists as the source of the virtual world but not as a god that protects its daily life. Once again, as happens in our understanding of magic and any metaphysical elements, this abstract power, this impersonal god does not belong to some other realm beyond the physical plane. He does not exist in a transcendental sphere that is separate from the physical plane. On the contrary, his seat of power lies on a higher plane within virtual reality, a plane that is loftier, inaccessible and untouchable, yet always physical.

On the other hand, in *Darksiders*, War (who is one of the Four Horsemen of the Apocalypse and the main character in the game), the Council, angels and demons are either on the side of evil because they clearly seek to destroy the divine world or incline towards evil because they are willing to kill and destroy and to play an active part in wars and power games (Xidakis 2018, pp. 248–60).

Therefore, the common element that links science fiction novels, comics, action and fantasy films and video games is the fact that all of the abovementioned powers and their like—gods, heroes, angels, demons, magicians and technologists—are, in accordance with the neomythological framework mentioned above, presented as natural, or rather supernatural, beings who, even though their abilities differ from those of ordinary men, are not transcendental in the sense of being something entirely different. This is why they are accepted by the plot, the readers, spectators and players as being credible characters.

This fact means that the powers that these beings possess, the manifestations of their divine, magical and supertechnological abilities, can be explained and can therefore happen. In other words, we have here not an incomprehensible form of magic but a form of science that is largely unknown.

We are thus dealing with a framework of "cosmotheism", a term used by the religious studies scholar Stelios Papalexandropoulos to describe the realm of New Religiosity, that is, with a practice involving the secularisation of the divine, and also with its identification and restriction to the realm of the physical world.

In the same way, in neomythology, and this is also clearly observed in diffused religiosity, the expressions of secular elements (a global energy), supernatural heroes (Superman, etc.), extra-terrestrial creatures (angels, gods, space beings, etc.), discussions about the existence of other dimensions (fourth dimension), a higher form (the inner self or the real person), their powers, and the magical and supernatural appearance of these beings and situations, undergo "scientification", as he termed it, i.e., they are explained in logical, "scientific" terms and they always identify with physical space and its elements (Papalexandropoulos 1992, pp. 50–51; 1997, 168–69; Hanegraaff 1999, p. 146; Clarke 2006, pp. 255, 257).

The result of this cosmotheism, of this integration of the divine and transcendental element in the material and explainable world, is the objectification of the divine. In other words, God, as placed exclusively within the boundaries of the cosmos, is treated as an object that will enhance a positive situation or create a negative one, which will help one person or harm another. As a result of this perspective, the supernatural—in the sense of magic—is treated as a tool (Xidakis 2018, pp. 315–16).

This is why in video games, which are essentially neomythological products, we similarly see gods serving as the main character's helper, as in the *Mana* series (Square, 1993) ("Mana" 2021); deities divided into levels of a greater or lesser nature, as in the *Xenosaga* series (Monolith, Namco Bandai, 2002 et seq.) ("Xenosaga" 2021); and generally dragons and crystals, temples and laboratories, and living planets identifying with the absolute.

The absolute is sometimes presented as supernatural, superior, stronger, purer and wiser, and is placed above the digital hero, who may be in a state of weakness, illness or even amnesia; at times it is presented as something material that shares the same world as the hero, and at other times, it is presented as a means to save the digital protagonist. Similarly, conquering the more powerful is presented not just as a means to an end but as an end in itself. The successful virtual hero who is controlled by their player/operator is

the one who can and knows how to upgrade themselves and to acquire super weapons and tools that will provide immediate solutions (Xidakis 2018, p. 316).

Within this cosmotheism, religion, as in *Deus Ex: Invisible War* (Eidos Interactive 2003), is presented as an organisation, as a multinational and oftentimes faceless corporation ("Deus Ex: Invisible War" 2022). On the other hand, in video game plots, magic is identified with science, while in terms of actually playing the game, it is presented in a computational-statistical way. This means that the magical skills of a virtual character are presented through the script as explainable powers and are portrayed on screen with charts and numbers (Xidakis 2018).

Irving Hexham and Karla Poewe believe this connection between magic and technology is to be found in the sphere of diffused religiosity and this is why they speak of (a) a new mythology, the mythology of evolution, which, in effect, is presented by means of rules and tangible results, like science (Hexham and Poewe 1997, pp. 92, 93); (b) a new science, which is based on insight, magic and communication with other, non-human beings (Hexham and Poewe 1997, p. 93); and (c) a mythological-pseudoscientific sphere in which everything is possible: the human race can become god, through the process of evolution and upgrading, since the universe itself is undergoing the same process (Hexham and Poewe 1997, p. 95).

Similarly, Wouter J. Hanegraaff, in his study on the New Age, that new religious movement that appeared after the 1960s, also speaks of the "new science" and the "mythology of science". More specifically, he, in turn, also referred to the desire of New Ager to (a) question official science; (b) support the existence of a new science, one that will not only study the causal relations between natural phenomena but will also have the capacity to know even the very nature of the divine element in the world; (c) treat science just like men in previous centuries treated magic—as a sphere, that is, in which everything is interconnected (Hanegraaff 1996, pp. 62, 63, 398, 407, 421, 423, 517; 1999, pp. 153, 154), and in this way, logic can be used to explain (as if it were natural knowledge) not only the secrets of matter but also those of the spirit, of consciousness, or—as it is otherwise called in New Age terminology—the Self (Hanegraaff 1999, pp. 155, 156, 158).

The inclusion of the gods and superhumans in the supernatural and not the metaphysical sphere (which characterises God in the monotheistic religions) can thus be observed in the New Religiosity. This represents a tendency to secularise the divine, that is, to place it in a worldly context, to confine it within the bounds of the natural world. Sometimes obviously, for example in the UFO Religions (Xidakis 2006), and at other times more indirectly, as in the case of the Ascended Masters of the Theosophical Society (Xidakis 2006, pp. 20–24), those who in the teachings of the New Religions possess the status and role of the gods are placed in a higher sphere of reality. The same thing happens in the case of angels (Sutcliffe 2003, pp. 57, 75, 79, 86, 96, 145–46, 170; Hanegraaff 1996, p. 198) and the ancient gods (Hanegraaff 1996, p. 202) in the sphere of the New Age movement. In the same way, the manifestations of all these supernatural beings, their powers and their magical appearances are explained in terms of logic and "science" and are always identified with the natural world and its constitutive elements.

Hence, the neomythology that features such gods and supernatural heroes or antagonists also belongs to the New Religiosity and repeats motifs that are found in the teachings of the New Religions and in New Age myths.

## 5. Conclusions

In the plots of science fiction works and their characters, the mythical element predominates. It would be no exaggeration to say that these plots are products of mythology in the sense that they reproduce old mythological schemata, and in so doing, produce new myths. This reproduction and production of mythical elements, and in particular, the way in which these mythical elements are combined and presented, call to mind the more general phenomenon of neomythology and the framework it forms part of, namely the New Religiosity.

What common element may be observed in these two cases—the one specific and the other general—of neomythology and the New Religiosity? The common element is the sphere of myth, which has not only not become defunct within the confines of modern life but is in fact flourishing. In neomythology, traditional myths are presented in the plots of films, literary works and video games in a new way, and in the New Religiosity too, myths, either in the teachings and closed framework of a New Religion, or in the endless, casual and occasionally open discussions of New Agers, are utilised in a variety of different ways.

In this sphere of neomythology and the New Religiosity, there are five dominant motifs, as follows.

First of all, the placing of the divine in a worldly context, that is to say, the inclusion of the transcendental element in the material and explainable world, along with a derivative of this motif: the objectivisation of the divine, in other words, the treatment of gods as if they were objects that can improve positive situations or create negative ones, objects that can help certain people or harm others.

Second, the absolutisation not only of the acquisition of power but also those who bear it. The supporter of the New Religiosity and the fantastical hero is often presented as the person who has attained a state of self-actualisation, self-development and self-expression, and in all cases, the best image or version of themselves compared with all other uninitiated, ordinary people.

Third, a consequence of the two previous motifs is the correlation between magic and technology. The new religious adherent, depending on the teaching that follows and the group to which they belong, can be both a magician and a technologist at the same time, without either concept negating the other. In science fiction films and comics, as well as video games, both the good and bad virtual characters are often both magicians and technologists. Both of these attributes—those of magician and scientist—are reconciled and combined because they exist within the same framework in which the divine is to be found in material reality, though in a higher sphere or plane. Thus, for fantasy characters, magic is usually another version of super-knowledge or super-science, while science is another means of achieving what they seek in a sudden, spectacular fashion.

Fourth, a follower of the New Religiosity, like the characters in myth, can be the one who will receive the knowledge and the right to join a brotherhood of secret superhuman beings in a new age.

Fifth, in the New Religiosity the history of civilisation, and all that that encompasses—that is to say, myths, legends and traditions—becomes material that is used to narrate another, alternative side of it. This motif is a derivative of the previous ones since the new myth that is gradually being fashioned displays evidence of (a) the select nature of the followers of the New Religions, who know the new myth as a kind of esoteric knowledge and reproduce it; and (b) the inclusion of the absolute, that is to say, the divine, within the confines of the world, in the sense that our world is not the one we know but something else, which possesses something else, for example, another power that leads elsewhere, such as another, new age.

Similarly, in contemporary literary works, video games and cinema products, all of the mythical and religious elements—that is to say, the elements of the mythologies of the past and traditional religiosity—are present, although they are retold from the beginning.

In these literary works, video games and cinema products, then, myth is a dominant element and is developed either in accordance with the motifs of traditional mythology and fable, with respect to the hero and their adversary, or—and this is the most usual case—in accordance with the motifs of the new diffused religiosity and the broader framework of neomythology.

This is why in the plots of the abovementioned works, the roles of the good heroes and their evil enemies are played on the one hand by a variety of warriors and samurai and on the other by an array of aliens, angels, demons, higher and lesser gods, scientists who possess magical abilities and magicians that redefine themselves as scientists. All of

these characters might fight not only in scientific laboratories and modern cities but also in temples, basements and dark dungeons or magic forests and faraway planets.

The products of popular culture are, therefore, products of myth and produce myth themselves; they are inspired by myth and, what is more, by the mythological schemata found in the New Religiosity. In them, the modern Westerner witnesses the re-emergence of elements that they believed or thought or boasted that they no longer needed, such as (a) myth, (b) the distinction between good and evil, (c) the concept of the transcendental, and (d) religious and mythical symbols. All this ultimately causes us to reflect that modern man may still be a being that seeks and strives after myth and also reinforces it.

**Funding:** This research received no external funding.

**Conflicts of Interest:** The authors declare no conflict of interest.

## Notes

1  A preliminary and brief study of this subject, which was carried out as part of the initial research into the phenomenon, was presented in Greek in the theological journal Synaxis (Xidakis 2019).

2  In order to have a better and safer view of the plots, we consulted the wikia and wikipedia websites, to which the international community of loyal fans of neomythology products, such as video games, has anonymously, but with constant updates and dedication, provided comprehensive information on the scripts of the fantasy movies, sci-fi series and video games. Of course, where secondary literature sheds more light on certain parameters of these scripts, a comparison has been made between these references and those provided by the above websites.

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
