# Peer review of "Neomythology: A New Religious Mythology"

_religions, doi:10.3390/rel13060536_

Round 1
Reviewer 1 Report
This is an interesting and insightful article. I would recommend minor changes to give a stronger sense of introduction (I think the reader is thrust into a discussion of myth before we get an understanding of what the article sets out to achieve). The conceptual links across different sci-fi texts could perhaps be highlighted in a more systematic way so that it doesn't risk coming across as a meandering description of different films and video games - I don't think it quite comes across as this, but I would certainly recommend tightening the discussion to bring forth the argument more. The link between secularisation of the divine and objectification of the divine (470-482) I found particularly interesting, but the ways that the latter follows on from the former are not fully explicated and thus the link is unclear. It also does not appear in the conclusion, so I wonder about its relevance to the discussion and overarching argument? The only other fairly minor comment was about the gendered language to refer to 'man' - it strikes me as unnecessary and not serving the analysis in any way.
Author Response
I am forwarding a file with the changes that have been made to the article based on your comments.

Reviewer 2 Report
The research is very interesting and starts very well with the clarification of the theoretical bases of the work. However, when you get to section 4, dedicated to the distinctive characteristics of neomythology, I think you should clarify in the title of that section in the cinema, the comics... On the other hand, this section, which precedes the conclusions, is messy and should be structured by the five motifs that appear in the conclusion. In this way, said conclusion would be related to the analysis of the selected works. On the other hand, the research does not delimit the scope of neomythology in the field of media such as cinema, comics, etc. and in everyday reality. Rather, the author hindered the collective imaginary that these means of expression build. Finally, it would not hurt if the author reflected on the role played by the products of popular culture in the recreation and production of myths.
Author Response

(The authors gave the same response as above.)

Round 2
Reviewer 2 Report
The author has made the indicated changes. Now his work is ready to be published